# Antarctic Krill Oil Ameliorates Monosodium Iodoacetate-Induced Irregularities in Articular Cartilage and Inflammatory Response in the Rat Models of Osteoarthritis

**DOI:** 10.3390/nu12113550

**Published:** 2020-11-20

**Authors:** Minhee Lee, Dakyung Kim, Soo-Jeung Park, Jeong moon Yun, Dong Hwan Oh, Jeongmin Lee

**Affiliations:** 1Department of Medical Nutrition, Kyung Hee University, Yongin 17104, Korea; miniclsrn@khu.ac.kr (M.L.); k4kyung@naver.com (D.K.); sujeungp@khu.ac.kr (S.-J.P.); moon1894@hanmail.net (J.m.Y.); treeodong96@naver.com (D.H.O.); 2Research Institute of Clinical Nutrition, Kyung Hee University, Seoul 02247, Korea

**Keywords:** Antarctic krill oil, monoiodoacetate, osteoarthritis

## Abstract

The aim of this study was to examine the effects of Antarctic krill oil (FJH-KO) in a rat model of monosodium iodoacetate (MIA) induced osteoarthritis. The effect of FJH-KO on the development and severity of MIA-induced osteoarthritis was assessed using hematoxylin and eosin (H&E) staining and micro-CT. The expression of PGE_2_, pro-inflammatory cytokines (IL-1β, TNF-α), and arthritics related genes in osteoarthritic rats in response to FJH-KO supplementation was investigated using real time PCR. FJH-KO supplementation in the arthritic rat model reduced tissue damage, cartilage degeneration, and reduced the MIA-induced irregularities in articular cartilage surface. Serum PGE_2_, IL-1β, IL-6, and TNF-α levels were higher in MIA treated animals, but these levels decreased upon FJH-KO supplementation. When FJH-KO was provided at a dose of 150 mg/kg b.w to MIA-treated animals, it significantly increased the mRNA expression of anabolic factors. The mRNA expression of catabolic factors was significantly decreased MIA-treated animals that were provided FJH-KO at a dose of 100 and 150 mg/kg b.w. Moreover, the mRNA expression of inflammatory mediators was significantly decreased MIA-treated animals supplemented with FJH-KO. These results suggest supplementation with FJH-KO ameliorates the irregularities in articular cartilage surface and improves the inflammatory response in the osteoarthritis. Thus, FJH-KO could serve as a potential therapeutic agent for osteoarthritis treatment.

## 1. Introduction

Osteoarthritis clinically characterized by progressive loss of articular cartilage is the most commonly encountered arthropathy in the elderly. This condition affects the joints that are regularly used and joints that support most of the weight [1,2]. Under normal conditions, chondrocytes respond to tissue injury via enhanced proteoglycan and collagen synthesis. However, when repair fails, articular cartilage degeneration occurs [3,4]. Arthritis is caused when there is an imbalance between the degradation and synthesis of articular cartilage. Processes associated with the degradation of articular cartilage result in the production of pro-inflammatory including IL-1β, TNF-α, and catabolic mediators including matrix metalloproteinases (MMP)-3, MMP-7, and MMP-13 [4,5]. 

Osteoarthritis treatment is aimed at the following: reducing pain, delaying disease progression, relieving disease, improving or maintain functional status, and minimizing cartilage damage [6]. The general recommended treatments for osteoarthritis include aerobic exercise, and intervention including corticosteroids, acetaminophen, or non-steroidal anti-inflammatory drugs (NSAIDs) that are intended to reduce pain and inflammation and inhibiting disease [7,8]. However, such mediation are associated severe adverse effect, such as renal toxicity, diarrhea, vomiting, gastrointestinal disturbances, nausea, or increased cardiovascular risks [9]. Recent studies have investigated the use of various complementary and alternative approaches including healthy functional food, nutraceuticals, and dietary supplements to manage osteoarthritis delaying disease progression and reliving pain [10,11]. 

Therefore, we investigate the activity and effects of Antarctic krill oil (FHJ-KO) on monosodium iodoacetate (MIA)-induced osteoarthritis in a rat model. Krill oil (KO) is a safe to eat oil extracted from krill that contains long-chain n-3 polyunsaturated fatty acids (PUFAs) such as phospholid-type eicosapentaenoic acid and phospholipid type decosahexanoic acid [12]. N-3 PUFA has been reported to exert anti-tumorigenic, anti-inflammatory effects [13]. In this study, we investigated the effect of FHJ-KO on structural changes, anabolic and catabolic factors, and inflammatory responses in rat model of MIA-induced osteoarthritis.

## 2. Materials and Methods 

### 2.1. Preparation of the Extract

The Antarctic krill oil (FJH-KO) was supplied from Frombio (Suwon-si, Gyeonggi-do, Republic of Korea). The extracts was stored at −20 °C until used. 

### 2.2. Animal Treatment and MIA Induced OA in Rat 

The experimental protocol was approved by the Animal care and Use Review Committee of Kyung Hee University (KHGASP-19-331). Male Sprague-Dawley rats (6 weeks old) were obtained from Japan SLC, Inc., Shizuoka, Japan and were housed in a standard environment (12 h light/dark cycle, 22 ± 2 °C; and 50–60% humidity). The rats were randomly allotted into six groups of six rats each; normal AIN 93G diet (normal control; NC), osteoarthritis induction and normal AIN 93G diet (control; C), osteoarthritis induction and AIN 93G diet containing methylsulfonylmethane (MSM) 150 mg/kg b.w. (positive control; PC), osteoarthritis induction and AIN 93G diet containing FJH-KO 50 mg/kg b.w. (FJH-KO50), osteoarthritis induction and AIN 93G diet containing FJH-KO100 mg/kg b.w. (FJH-KO100), osteoarthritis induction and AIN 93G diet containing FJH-KO150 mg/kg b.w. (FJH-KO150). MIA was injected into the right knee joint after 7 days dietary administration. The rats were euthanized on day 31, and knee joint tissues and serum were collected for cytokines and mRNA expression analysis.

### 2.3. Histologic Staining and Micro-Computed Tomography (μCT) Image Scan

The knee joints of the rats were removed, fixed using 10% formalin, and decalcified, embedded in paraffin, and sectioned 7 μm. The sections were stained with hematoxylin and eosin (H&E) for histological assessment. 

Bone mineral density (BMD), bone volume/total tissue volume, trabecular number, trabecular thickness, and trabecular separation were evaluated using the Skyscan 1172 X-ray μCT scanning system (Bruker, Belgium). μCT images were acquired using formalin-fixed articular cartilage.

### 2.4. Measurement of Serum PGE_2_ and Cytokines Levles

The levels of PGE_2_, interleukin-1β (IL-1β), and TNF-α were measured using an ELISA kit (R&D system, Minneapolis, MN, USA).

### 2.5. mRNA Expression in Rat Cartilage

mRNA were extracted from the rat articular cartilage using the RNeasy Mini Kit (QIAGEN, Germantown, MD, USA). cDNA was synthesized using iScript™ cDNA Synthesis kit (BIORAD, Hercules, CA, USA). Real-time polymerase chain reaction (RT-PCR) was performed using SYBR Green PCR Master Mix (iQ SYBR Green Supermix, BIORAD). cDNA was used as template for RT qPCR on a CFX Connect™ real-time PCR detection system (BIORAD) using a two-step method (95 °C for 30 s, 56 °C for 30 s, 72 °C for 45 s) for 40 cycles. Primer pairs are presented in Table 1. Data analysis was performed using the CFX Maestro^TM^ Analysis Software (BIORAD).

### 2.6. Statistical Analysis

All results were expressed as mean ± standard deviation (SD). Statistical analysis was performed using one-way ANOVA with SPSS SPSS PASW Statistic 23.0 (SPSS Inc., Chicago, IL, USA). Duncan’s multiple range test was used to examine the differences among the groups and a *p* < 0.05 was considered significant.

## 3. Results

### 3.1. Histological Analysis of the Articular Cartilage

Histological analysis of hematoxylin/eosin-stained knee sections after intra-articular injection of MIA revealed cartilage degeneration and irregular articular cartilage surface in osteoarthritic rats. In rats, tissues section of FJHKO supplemented osteoarthritic rats revealed reduced histological damage and cartilage degeneration, and less irregular articular cartilage surface (*p* < 0.05) (Figure 1).

### 3.2. FJH-KO Supplemenatation Improved Mineralization Parameters

We measured the bone mineral density (BMD), bone volume/total tissue volume (BV/TV), trabecular number, thickness, and separation in cortical bone. BMD, BV/TV, Th.N, and Tb.Th were decreased in rats with MIAosteoarthritis. FJH-KO supplementation increased the BV/TV, Th.N, Tb.Th more than MIA-induced rats. Tb.Sp was increased in rats with MIA-induced osteoarthritis. FJH-KO supplementation in these rats decreased the Tb.SP (*p* < 0.05) (Figure 2, Table 2).

### 3.3. FJH-KO Supplementation Reduced Serum PGE_2_ and Pro-Inflammatory Cytokines Levels 

Pro-inflammatory cytokine plays a key role in the development and maintenance of chronic inflammation and PGE_2_ secretion [14].

We analyzed the levels of PGE_2_ and pro-inflammatory cytokines in the serum. The serum PGE2, IL-1β, IL-6, and TNF-α levels were higher in rats with MIA induced osteoarthritis; however, these levels were suppressed in FJH-KO supplemented rats with MIA-induced osteoarthritis compared with those in rats with MIA-induced osteoarthritis (*p* < 0.05) (Figure 3). 

### 3.4. FJH-KO Supplementation Ameliorated mRNA Expression of Anabolic and Catabolic Factors in the Articular Cartilage

We analyzed the expression of anabolic and catabolic factors at the mRNA level in the articular cartilage to determine the effect of FJH-KO on osteoarthritis. The mRNA expression of anabolic factors such as aggrecan, collagen type I and type X, tissue inhibitor of metalloprotease (TIMP)-1, and TIMP-3 was significantly increased in rats in FJH-KO 150 group compared with that in rats with MIA induced osteoarthritis, but the expression of collagen type II mRNA did not differ significantly among all the MIA-treated groups. (*p* < 0.05) (Figure 4). The mRNA expression of catabolic factors MMP-3, MMP-7, and MMP-13 was significantly decreased in rat in the FJH-KO100 and FJH-KO150 groups compared with that in rats with MIA induced osteoarthritis (*p* < 0.05) (Figure 4). Furthermore, the mRNA expression of inflammatory mediators COX-2, IL-1β, TNF-α, and NF-κB was significantly decreased in the FJH-KO supplemented rats compared with that in rats with MIA induced osteoarthritis (*p* < 0.05) (Figure 4).

### 3.5. FJH-KO Supplementation Ameliorated the mRNA Expression of Inflammatory Factors in the Articular Cartilage

We analyzed the expression of the inflammatory factors at the mRNA level in the articular cartilage to estimate the effect of FJH-KO on osteoarthritis. The mRNA expression of pro-inflammatory mediators IL-1β and TNF-α decreased significantly in the FJH-KO supplemented rats compared with in rats with MIA induced osteoarthritis (*p* < 0.05) (Figure 5). The mRNA expression of COX-2 and NF-κB was significantly decreased in the FJH-KO supplemented rats compared with in rats with MIA induced osteoarthritis (*p* < 0.05) (Figure 5). 

## 4. Discussion

Osteoarthritis is characterized by the formation of osteophytes, biochemical changes in the synovial membrane, and destruction of the articular cartilage, all of which lead to excess production of catabolic and pro-inflammatory mediators [15]. Anti-inflammatory therapeutic agents and NSAIDs have been shown to be effective for treatment of osteoarthritis. However, the use of these drugs is associated with adverse gastrointestinal effects such as heartburn, stomach pin, and ulcers [9]. Therefore, healthy functional food, nutraceuticals, and dietary supplements have recently emerged as an alternative strategy to treat osteoarthritis, as these interventions are associated with minimal adverse effects and toxicity. Therefore, we evaluated the effect of FJH-KO on a rat model of MIA-induced osteoarthritis. The MIA induces joint pain, articular cartilage degradation, and acute inflammation [16]. Our data show that MIA treatment induced matrix degradation and disrupted surface of the articular cartilage, resulting in the generation of an irregular surface, as revealed by histological analysis, and supplementation with FJH-KO in the this background reduced cartilage destruction. Previous studies have demonstrated that omega-3 and/or KO supplement reduced cartilage destruction and OARSI score [17,18,19].

Pro-inflammatory cytokines including IL-1β and TNF-α play an important role in osteoarthritis. These inflammatory mediators induce the degradation of the articular cartilage. Excessive PGE2 result in higher productionIL-6 [20]. Our data show that FJH-KO supplementation in rats with MIA-induced osteoarthritis significantly decreased the serum levels of PGE_2_, IL-1β, and TNF-α compared with those in rats without FJH-KO supplementation. Previous studies have reported an algesic and anti-inflammatory effects of KO in mouse model of carrageenan-induced inflammation [21,22]. 

Aggrecan is a proteoglycan that is an essential component of the extracellular matrix, and is therefore essential for the normal functioning of the joints and the formation of the articular cartilage [23]. Aggrecan expression been shown to be downregulated by IL-1β via the ERK and MAPK pathways in human chondrocyte [24,25]. Collagen is one of components of the extracellular matrix in healthy conditions; collagen type II is mainly found in cartilage matrix. Significant down-regulation of collagen type II has been shown in a rat model of osteoarthritis [26,27]. MMPs are a family of zinc-dependent proteinase, which play a crucial role in various processes, including cancer progression, release of growth factors, and tumor angiogenesis. The activities of MMPs are regulated by TIMPs. Both MMPs and pro-inflammatory cytokines are expressed at high levels in the articular cartilage of osteoarthritis patients [28,29]. Our data show that FJH-KO supplementation in rats with MIA induced osteoarthritis significantly increased the expression of aggrecan and collagens compared with that in rats not supplemented with FJH-KO. We need to confirm the expression of MAPK and ERK pathway intermediaries at mRNA and protein levels in rats with MIA induced osteoarthritis to evaluate the molecular function of FJH-KO. Additionally, our data show that FJH-KO supplementation in rats with MIA induced osteoarthritis significantly decreased the expression of TIMPs and MMPs compared with that in rats without FJH-KO supplementation.

Pro-inflammatory cytokines induce the destruction of articular cartilage and degradation of the cartilage matrix, thereby causing collagen and proteoglycans less. IL-1β and TNF-α were shown to increase the expression of MMPs and decrease the expression of proteoglycans and collagen. Pro-inflammatory cytokines enhance the activities of COX-2 and NF-κB [30]. Our data showed that FJHKO supplementation in rats with MIA induced osteoarthritis significantly decreased the expression of pro-inflammatory cytokines (IL-1β, TNF-α) and inflammatory factors compared with that in rats without FJH-KO supplementation. However, we need to confirm the expression of NF-κB pathway intermediaries at the protein level in rat with MIA-induced osteoarthritis to evaluate the molecular function of KO. 

We evaluated the anti-inflammatory effects of FJH-KO on MIA-induced osteoarthritis in rats by measuring the level of pro-inflammatory cytokines, and found that FJH-KO supplementation reduced the levels of cytokines. Further, we also found that FJH-KO supplementation reduced the expression of MMPs, and inflammatory mediators (IL-1β, TNF-α, COX-2, and NF-κB).). In a study performed by Wang et al., KO inhibited cartilage degeneration and maintained the normal chondrogenic phenotype in destabilization of the medial meniscus mouse, by regulating autophagy and apoptosis [31]. In addition, they also reported that peptide from KO improved osteoarthritis via inhibition of HIF-2α-mediated death receptor apoptosis and -inflammatory cytokines expression [32]. Park et al. showed that a mixture of krill oil, astaxanthin, and hyaluronic acid reduced serum levels of the pro-inflammatory cytokines as well as mRNA expression levels of iNOS and COX-2 in the knee joint from MIA-induced osteoarthritis [17]. These previous results and our previous results suggest that suggest that KO may reduce the inflammatory response and improve clinical symptoms of osteoarthritis.

## 5. Conclusions

In conclusion, we investigated the effect of FJH-KO in rat models of MIA-induced osteoarthritis. We found that the anti-osteoarthritic effects of FJH-KO were associated with the protection of articular cartilage and not with inflammation and degradation through the downregulation of pro-inflammatory cytokines. Therefore, we suggest that FJH-KO could serve as a potential therapeutic agent for osteoarthritis treatment. 

## Figures and Tables

**Figure 1 nutrients-12-03550-f001:**
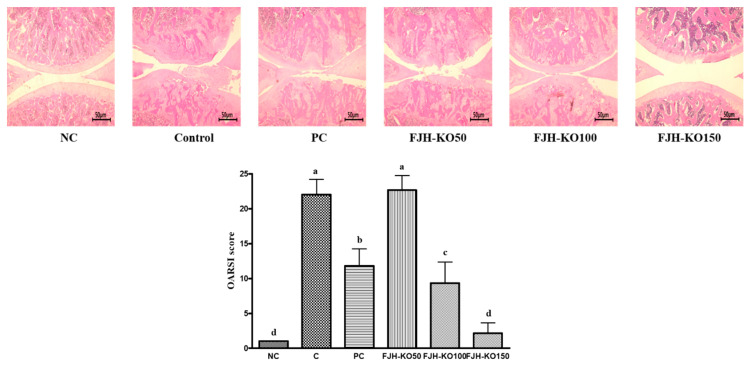
H&E staining in the knee joints of normal rats and osteoarthritic rats. NC: AIN93G diet; C: AIN93G diet + MIA injected group; PC: AIN93G diet + MIA injected group with supplemented 150 mg/kg b.w MSM; FJH-KO 50: AIN93G diet + MIA injected group with supplemented 50 mg/kg b.w KO; FJH-KO 100: AIN93G diet + MIA injected group with supplemented 100 mg/kg b.w FJH-KO; FJH-KO 150: AIN93G diet + MIA injected group with supplemented 150 mg/kg b.w FJH-KO. Values are presented as means ± SD values. Different letters show a significantly difference at *p* < 0.05 as determined by Duncan’s multiple range test.

**Figure 2 nutrients-12-03550-f002:**
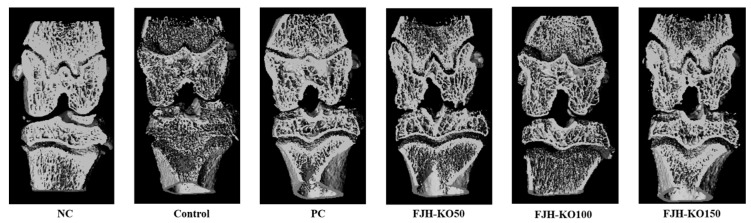
μCT analysis of the hind knee joint rats with or without MIANC: AIN93G diet; C: AIN93G diet + MIA injected group; PC: AIN93G diet + MIA injected group with supplemented 150 mg/kg b.w MSM; FJH-KO 50: AIN93G diet + MIA injected group with supplemented 50 mg/kg b.w KO; FJH-KO 100: AIN93G diet + MIA injected group with supplemented 100 mg/kg b.w FJH-KO; FJH-KO 150: AIN93G diet + MIA injected group with supplemented 150 mg/kg b.w FJH-KO

**Figure 3 nutrients-12-03550-f003:**
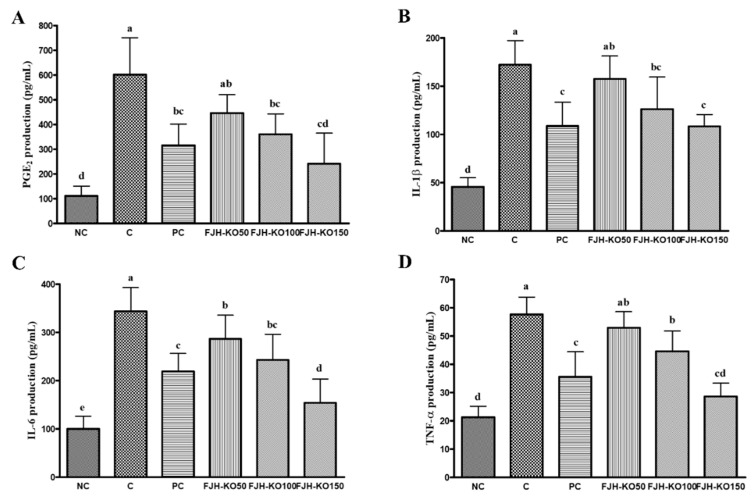
Effect of krill oil on serum PGE_2_ and pro-inflammatory cytokines levels in rat model with or without MIA-induced osteoarthritis. (**A**) PGE_2_, (**B**) IL-1β, (**C**) IL-6, (**D**) TNF-α. NC: AIN93G diet; C: AIN93G diet + MIA injected group; PC: AIN93G diet + MIA injected group with supplemented 150 mg/kg b.w MSM; FJH-KO 50: AIN93G diet + MIA injected group with supplemented 50 mg/kg b.w FJH-KO; FJH-KO 100: AIN93G diet + MIA injected group with supplemented 100 mg/kg b.w FJH-KO; FJH-KO 150: AIN93G diet + MIA injected group with supplemented 150 mg/kg b.w FJH-KO. Values are presented as means ± SD values. Different letters show a significantly difference at *p* < 0.05 as determined by Duncan’s multiple range test.

**Figure 4 nutrients-12-03550-f004:**
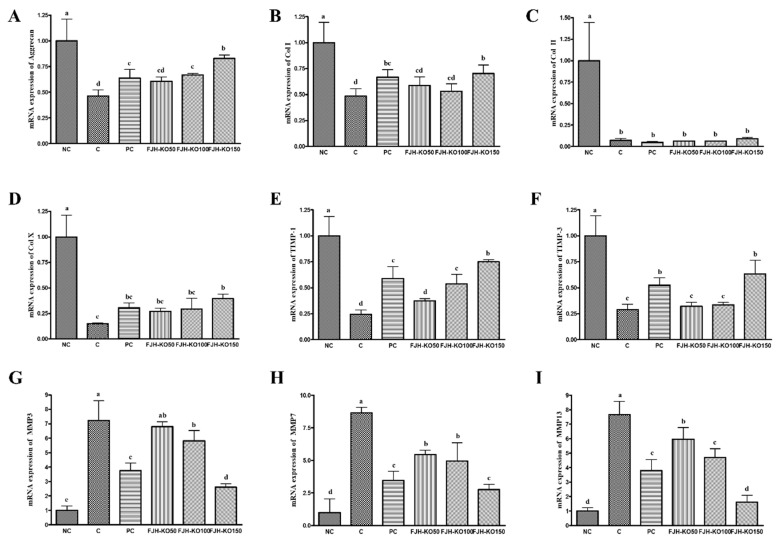
Effect of krill oil on mRNA expression of anabolic factors and catablic factors in rat model with or without induced osteoarthritis by MIA. (**A**) aggrecan, (**B**) Collagen type I, (**C**) Collagen type II, (**D**) Collagen type X, (**E**) TIMP-1, (**F**) TIMP-3, (**G**) MMP3, (**H**) MMP7, (**I**) MMP13. NC: AIN93G diet; C: AIN93G diet + MIA injected group; PC: AIN93G diet + MIA injected group with supplemented 150 mg/kg b.w MSM; FJH-KO 50: AIN93G diet + MIA injected group with supplemented 50 mg/kg b.w FJH-KO; FJH-KO 100: AIN93G diet + MIA injected group with supplemented 100 mg/kg b.w FJH-KO; FJH-KO 150: AIN93G diet + MIA injected group with supplemented 150 mg/kg b.w FJH-KO. Values are presented as means ± SD values. Different letters show a significantly difference at *p* < 0.05 as determined by Duncan’s multiple range test.

**Figure 5 nutrients-12-03550-f005:**
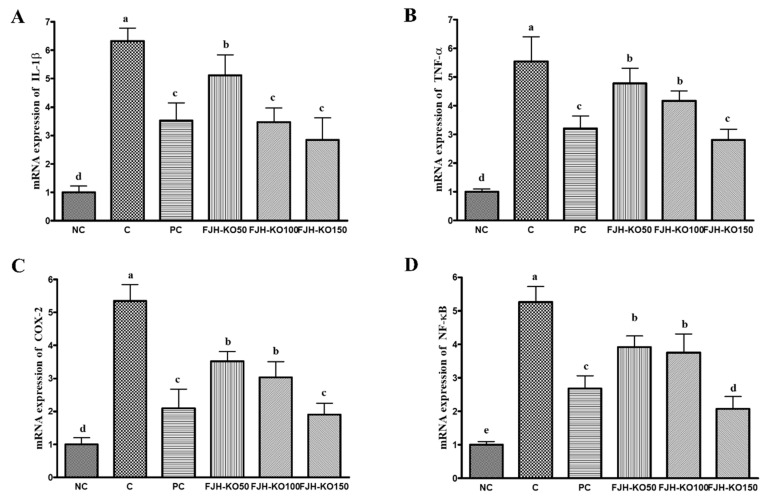
Effect of krill oil on mRNA expression of inflammatory mediators in rat model with or without induced osteoarthritis by MIA. (**A**) IL-1β, (**B**) TNF-α, (**C**) COX-2, (**D**) NF-κB. NC: AIN93G diet; C: AIN93G diet + MIA injected group; PC: AIN93G diet + MIA injected group with supplemented 150 mg/kg b.w MSM; FJH-KO 50: AIN93G diet + MIA injected group with supplemented 50 mg/kg b.w FJH-KO; FJH-KO 100: AIN93G diet + MIA injected group with supplemented 100 mg/kg b.w FJH-KO; FJH-KO 150: AIN93G diet + MIA injected group with supplemented 150 mg/kg b.w FJH-KO. Values are presented as means ± SD values. Different letters show a significantly difference at *p* < 0.05 as determined by Duncan’s multiple range test.

**Table 1 nutrients-12-03550-t001:** Primer set sequence used for Real-Time PCR.

Sequence Name	Sequence Number	Primer Sequences
GAPDH (rats)	NM_017008	F 5′-TGG CCT CCA AGG AGT AAG AAA C-3′ R 5′-CAG CAA CTG AGG GCC TCT CT-3′
Aggrecan (rats)	NM_022190	F 5′-GAA GTG TCC AAA CCA A-3′ R 5′-CGT TCC ATT CAC CCC TCT CA-3′
Col I (rats)	NM_000088	F 5′-GAG CGG AGA GTA CTG GAT CGA-3′ R 5′-CTG ACC TGT CTC CAT GTT GCA-3′
Col II (rat)	L48440.1	F 5′-GCA ACA GCA GGT TCA CGT ACA-3′ R 5′-TCG GTA CTC GAT GAT GGT CTT G-3′
Col X (rats)	XM_001053056	F 5′-TTC AGG GAG CGC GAT CAT-3′ R 5′-GAG GAG TAG AGG CCG TTC GAT-3′
TIMP-1 (rats)	NM_053819	F 5′-AAG GGC TAC CAG AGC GAT CA-3′ R 5′-ATC GAG ACC CCA AGG TAT TGC-3′
TIMP-3 (rats)	RNU27201	F 5′-GAC CGA CAT GCT TC CAA TTT C-3′ R 5′-GCT GCA GTA GCC ACC CTT CT-3′
MMP-3 (rats)	NM_133523	F 5′-GAG TGT GGA TTC TGC CAT TGA G-3′ R 5′-TTA CAG CCT CTC CTT CAG AGA-3′
MMP-7 (rats)	NM_012864	F 5′-ACT CTA GGC CAT GCC TTT GC-3′ R 5′-CCA TCC GTC CAG TAC TCA TCC-3′
MMP-13 (rats)	NM_133530.1	F 5′-ACG TTC AAG GAA TCC AGT CTC-3′ R 5′-GGA TAG GGC TGG GTC ACA CTT-3′
COX-2 (rats)	S67722	F 5′-AGA GAA AGA AAT GGC TGC AGA GTT-3′ R 5′-AGC AGG GCG GGA TAC AGT-3′
IL-1β (rats)	NM_031512	F 5′-GGC TTC GAG ATG AAC AAC AAA AA-3′ R 5′-GTC CAT TGA GGT GGA GAG CTT T-3′
TNF-α (rats)	AJ002278	F 5′-ACA AGG CTG CCC CGA CTA T-3′ R 5′-CTC CTG GTA TGA AGT GGC AAA TC-3′
NF-κB (rats)	NM_001276711	F 5′-GCA CCA AGA CCG AAG CAA TT-3′ R 5′-GAA ACC CCA CAT CCT CCT CT T-3′

**Table 2 nutrients-12-03550-t002:** Morphological parameters of cortical bone.

Measurements	NC ^1^	Induced Arthritis				
		C	PC	FJH-KO 50	FJH-KO 100	FJH-KO 150
BMD ^2^	796.1 ± 17.3 ^a^	620.0 ± 25.4 ^b^	700.3 ± 13.1 ^b^	621.5 ± 16.0 ^b^	639.6 ± 6.6 ^b^	675.8 ± 66.8 ^b^
BV/TV	0.30 ± 0.03 ^a^	0.14 ± 0.04 ^c^	0.26 ± 0.05 ^ab^	0.20 ± 0.07 ^bc^	0.22 ± 0.04 ^ab^	0.23 ± 0.03 ^ab^
Th.N	1.44 ± 0.02 ^a^	1.13 ± 0.04 ^b^	1.33 ± 0.04 ^ab^	1.21 ± 0.20 ^b^	1.22 ± 0.09 ^b^	1.32 ± 0.01 ^ab^
Tb.Th	0.25 ± 0.00 ^a^	0.11 ± 0.01 ^c^	0.19 ± 0.00 ^ab^	0.16 ± 0.03 ^b^	0.17 ± 0.02 ^b^	0.18 ± 0.02 ^b^
Tb.Sp	0.52 ± 0.02 ^b^	0.64 ± 0.02 ^a^	0.54 ± 0.08 ^b^	0.60 ± 0.09 ^ab^	0.58 ± 0.00 ^ab^	0.56 ± 0.01 ^ab^

^1^ NC: AIN93G diet; C: AIN93G diet + MIA injected group; PC: AIN93G diet + MIA injected group with supplemented 150 mg/kg b.w MSM; FJH-KO 50: AIN93G diet + MIA injected group with supplemented 50 mg/kg b.w FJH-KO; FJH-KO 100: AIN93G diet + MIA injected group with supplemented 100 mg/kg b.w FJH-KO; FJH-KO 150: AIN93G diet + MIA injected group with supplemented 150 mg/kg b.w FJH-KO. ^2^ BMD: bone mineral density; BV/TV: bone volume over total volume; Th.N: Trabecular number is the inverse of the mean spacing between the midlines of the trabeculae; Tb.Th: Trabecular thickness is the mean trabecular bone diameter; Tb.Sp: Trabecular separation is the mean distance between the borders of the segmented trabeculae. Values are presented as means ± SD values. Different letters show a significantly difference at *p* < 0.05 as determined by Duncan’s multiple range test.

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
