# Peer review of "Antarctic Krill Oil Ameliorates Monosodium Iodoacetate-Induced Irregularities in Articular Cartilage and Inflammatory Response in the Rat Models of Osteoarthritis"

_nutrients, 2020, doi:10.3390/nu12113550_

Round 1
Reviewer 1 Report
Dear Authors;
Thanks for your contribution; I have appreciated the manuscript but I consider that there are some issues that should be resolved.
Shortly:
1) No sample size a priori calculation about the number of rats used respect to the outcome expected.
2) Krill Oil appears to be a kind of "black box". I suggest to describe the chemical oil composition used in the experiment; the source (company, "home made or so on)
3) Histology doesnt give any information and pictures about is are not clear or useful to the reader. I suggest to use an OARSI score for rat OA.
4) I have appreciated the use of MicroCT but I would like to know exactly to priori as it's deisgned the VOI (volume of interest) considered in the specimen analyzed.
5) introduction and general conclusions should be integrated with some recent and significative papers that result neglected in the manuscript .
I would suggest to the Editors to consider these as minor revisions and if fulfilled the manuscript should be published.
best regards
Author Response
I appreciate your help on this manuscript and I thank you for your valuable comments about our laboratory work. We agreed to your comments and changed the manuscript as the followings.
- Thank you for your comments. We write that we used number of rats
- We explain that Krill oil extract method.
- As you requested in figure 1, We added figure 1 to supplement the information on the OARSI score.
- As you requested in MicroCT, we added the information about VOI in legend of figure 2.
- As you requested in discussion, we added discussion more.

Reviewer 2 Report
Dear authors,
Thank you for this study showing a possible benefit of krill oil in mice with OA.
- non-steroidal anti-inflammatory drugs (NASAIDs) --> NSAIDs
- Was MIA injected in all rats?
- Table 2 is difficult to interpret. I would prefer to clearly see what differs with either healthy control or control as now there is a lot of a`s and b`s without knowing what these p-value significance actually means. Same for fig 3/4/5.
- Why did you use MSM as positive control? This is quite crucial as the magnitude of this response is the comparator of the response of the intervention.
- Discussion feels like an elangotaed background. I would like to see more a discussion of the results.
Author Response
Thanks for your suggestions. We agreed to your comments and changed the manuscript as the followings.
Major points:
- Thank you for your comments. We changed NASAIDs-> NSAIDs
- As you requested MIA injection, we already wrote legend of all data. (MIA injection except for the normal diet)
- in figure 2, we replaced figure 2C and 2D with a more clearly visible band. Furthermore, in case of figure 2E, we added actin band, and present figure 2F to supplement the information on the expression of HSL.
- MSM is well known as a health functional food for joint health, and is a product that has been approved as a health functional food in Korea. Therefore, we used MSM as positive control.
- Thank you for your comments. We added discussion

Round 2
Reviewer 2 Report
Dear authors,
text is improved enormously. Good job. However, I find the figures still hard to understand. I would try not to use letters on all columns in fig4/5 but limit yourself to the stat sign differences between the most important groups.
Good luck!
Author Response
I appreciate your help on this manuscript and I thank you for your valuable comments about our laboratory work. We agreed to your comments and changed the manuscript as the followings. Thank you for your comments. I ran a statistical method through ANOVA rather than t-test to check the difference between all groups.
